# Patterns and Influencing Factors of Express Outlets in China

Xin Li  and Peng Zhang *

College of Geography Science, Harbin Normal University, Harbin 150025, China; hsd_lixin@163.com
* Correspondence: zhangp575@nenu.edu.cn

**Abstract:** China's express delivery industry has developed rapidly in the past decade, and the spatial distribution of express delivery outlets can reflect variations in regional development to a certain extent. Previous studies lacked point of interest (POI) data as the research object to analyze the limitations of express delivery outlets, and also lack a focus on the patterns that specifically occur in China. This study implemented spatial analysis methods such as nearest neighbor index, kernel density, and exploratory spatial data analysis to explore the spatial distribution patterns and characteristics of China's express outlets, and used geographic detector factor detection to objectively analyze the factors that affect their distribution. The following are the main conclusions of this study: (1) in China, express delivery outlets are more abundant in the east and sparser in the west; (2) the local agglomeration of express outlets presents a concentric structure, and outlier clusters appeared in the northeast and central regions; (3) the distribution of the express outlets resulted from the combined action of multiple factors (e.g., population factors, etc.). Our study not only explores and analyzes the overall situation in China but also broadens the scope of express outlet research.

**Keywords:** express outlets; spatial distribution pattern; influencing factors; China

## 1. Introduction

The express delivery industry is a fundamental, strategic, and livelihood-oriented industry in China and plays a crucial role in the development of the national economy. With the recent changes and upgrades in the consumption structure of Chinese urban and rural residents, the express delivery industry has recently become a key component of the macroeconomy. According to recent statistics, each person in China will receive and dispatch an average of 60 express parcels per year by the end of 2020, and express delivery businesses account for more than 60% of the global express parcel market. From a global perspective, China's express delivery industry has ranked first in the world for seven consecutive years [1]. The establishment of the modern express delivery industry in China can be traced back to 1896, when Emperor Guangxu of the Qing Dynasty officially approved the establishment of the Qing Post Office, which marked the birth of the modern China Post. However, it was not until the 1980s that international express delivery companies such as DHL and UPS flooded the Chinese market. Soon after, early domestic express delivery companies such as Datong Express and China Railway Express were established one after another. From the 1990s to the early 2000s, China's economy developed rapidly. Economic development, international and domestic economic and trade activities, and people's living needs provided good market conditions for the development of the express delivery industry. Express delivery companies such as Zhongtong, Yunda, and Yuantong were successively established during this period, and China's express delivery industry developed rapidly thereafter.

The earliest studies on the express industry were conducted in the United States. In the 1930s, Wayne L. McMillen conducted research on the American air express industry, which was the earliest article involving the express industry. The article summarized the characteristics, competitiveness, and future development of the air express industry [2]. However, few studies have been conducted on express network design, express market

supervision, market access and review, and the prospects of the express delivery industry. Goodovitch conducted a quantitative analysis on the location selection and optimization of an express delivery network in the early stages of the development of the express delivery industry [3]. Elhedhli took FedEx as an example and studied this company from an end network optimization perspective based on the principle of operations research [4] and analysis of competition, and put forward countermeasures and suggestions from a regulation perspective [5]. Many studies abroad have recently focused on express delivery research [6–8], including the use of drones for delivery [9] and the quality of express delivery services [10]. The express delivery industry has a relatively long history in western countries and there is a large amount of informatization in this field. However, very few studies have characterized the spatial distribution of express delivery outlets.

Research on the express delivery industry in China has remained largely descriptive, focusing on macroeconomics and management, transportation economics, and computer technology. After the reform and the opening up period, scholars have identified the difficulties faced by the development of China's express delivery industry, as well as supply and demand problems that need to be urgently solved [11]. During this time, China's express delivery industry grew rapidly and achieved great breakthroughs in operation, management, and technology [12–14]. After 2010, the e-commerce industry entered a stage of rapid development. As a new type of business, its distribution and related services have put forward many new requirements for the express delivery industry [15]. Nevertheless, because the express delivery industry in China is still in its infancy, this sector faces a plethora of challenges, including poor service, slow information update speeds, weak oversight from service providers and managers, and a general lack of talent across the entire industry [16]. Therefore, the development of strategies to improve the express delivery and e-commerce industries has recently garnered increasing attention among the scientific community [17,18]. As early as 2012, geographers have conducted research on the distribution and spatial organization of express delivery industry outlets at a spatial level. Taking SF Express [19] and Yuantong Express [20] as examples to explore the spatial organization of Chinese express delivery companies, a recent study found that cities with higher administrative levels exhibited higher network coverage rates. Furthermore, in cities with the same administrative level, the network coverage rate decreased gradually from south to north, which generally correlated with socioeconomic and population factors related to regional topography, local conditions, and industrial characteristics. Other studies have assessed express delivery volume [21], network elements of express [22], and express delivery service [23], and used ArcGIS and other approaches to study their spatial patterns. "Big data" research has been steadily increasing in recent years. In the field of express delivery research, some studies based on point of interest (POI) data have analyzed the spatial patterns of express delivery points and logistics points in Wuhan [24], Shenzhen [25], Chongqing [26], and other cities. However, the exploration of spatial patterns in the aforementioned studies lacks a comprehensive analysis from different scales. Additionally, there is a general lack of objective research methods for the exploration of influencing factors.

In terms of research methods, scholars mostly discuss express delivery issues from a theoretical level. Moreover, the SERVQUAL model [27], factor analysis [28], fuzzy comprehensive evaluation [29], and other methods are often used to comprehensively evaluate the quality of express delivery services. Most studies use ArcGIS to evaluate the layout of express delivery outlets. Moreover, a series of spatial analysis methods [30] such as nearest neighbor index and standard deviation ellipse, are used in related studies. However, few studies have analyzed the factors affecting the layout of express outlets, and most of the existing studies are based on the results of previous research. Among currently published exploratory studies, many factors have been evaluated using geographic indicators in China.

In summary, research on China's express delivery industry has mostly focused on the theoretical analysis of the industry and express delivery companies. Different countries may

exhibit variations in the market share of express delivery companies, as well as their service level. The importance of the express delivery industry in the national economy continues to rise. In turn, this not only promotes consumption and stimulates employment, but also aids in the development of agriculture, manufacturing, and foreign trade. Additionally, the express delivery industry has strongly supported the country's poverty alleviation strategy by promoting rural-to-urban agricultural sales and reverse sales of industrial products. Therefore, this important emerging industry deserves widespread recognition. China has a vast territory, and therefore its natural conditions and socio-economic development status vary greatly on a regional basis. Therefore, from a geographical perspective, this study explored the spatial distribution patterns of China's existing express delivery outlets in different regions and the factors that define these distribution patterns. Our findings thus provide a basis for the development of macro-control strategies for local express delivery companies, as well as for the optimization of their number and layout.

In this study, by systematically constructing a database of Chinese express outlets using spatial analysis methods such as the nearest neighbor distance method, kernel density, standard deviation ellipse, and exploratory spatial data analysis, we explored the spatial distribution patterns of China's express outlets from a national perspective using geographic detection. By following this approach, influencing factors were detected, a theoretical analysis of the decisive influencing factors was conducted, and constructive opinions were provided based on our findings.

## 2. Data Sources, Study Area, and Research Method

### 2.1. Data Sources and Research Areas

The sample data in this article included 179,770 express delivery outlets across the country as of early November 2021, distributed in 296 prefecture-level cities across the country, including counties and towns under their jurisdiction. The data were obtained from the AutoNavi map open platform by calling the web service API interface, and the coordinates were corrected using Q-Gis. The spatial data was obtained from the National Basic Geographic Information Center (http://www.ngcc.cn, accessed on 15 February 2022) (i.e., the 1:4 million vector map database of China). GDP, total social retail sales, and other data were obtained from the "China Urban Statistical Yearbook" and the statistical yearbooks of various cities. Due to limitations in data availability and the consistency of statistical coverage, Sansha City, provincial and county-level cities, Xinjiang Altai and other regions, autonomous prefectures, leagues, Hong Kong, Macau, and Taiwan were not included in this study.

### 2.2. Research Methods

#### 2.2.1. Nearest Neighbor Index

According to the nearest neighbor distance index, three distribution modes can be identified: when R > 1, the actual nearest neighbor distance between points is greater than the theoretical nearest neighbor distance, indicating that the points are mutually exclusive and tend to be uniformly distributed in space; when R = 1, the distribution of points originates from a completely random pattern and belongs to a random distribution in space; when R < 1, the actual nearest neighbor distance between points is smaller than the theoretical nearest neighbor distance and tends to be distributed in space [31]. This study sought to use the nearest neighbor distance method to calculate the nearest neighbor index of express outlets at a national scale, analyze the spatial distribution pattern of express outlets, and explore the spatial distribution pattern of express outlets at different scales.

This study sought to use the nearest neighbor distance method to calculate the nearest neighbor index of express outlets at the national scale, thus allowing for the analysis and exploration of the spatial distribution pattern of express outlets.

### 2.2.2. Standard Deviation Ellipse and Kernel Density

This study uses geographic information analysis methods to explore the spatial distribution patterns of China's express outlets, mainly using standard deviation ellipse and kernel density estimation. Among these methods, the standard deviation ellipse was determined by calculating the standard deviation of the x and y coordinates of the express distribution center, and an ellipse was constructed to obtain the position of the center of gravity and the distribution direction of the express outlets [32].

Kernel density analysis is a non-parametric estimation method implemented through the kernel density tool in the ArcGIS spatial analysis module. The calculation formula is as follows [33]:

$$f_n(x) = \frac{1}{nh^2\pi} \sum_{i=1}^{n} K\left(1 - \frac{(x-x_i)^2 + (y-y_i)^2}{h^2}\right)^2 \tag{1}$$

where $f_n(x)$ is the sum density value $K$ of a point in the study area; $x$ is the spatial weight function; $h$ is the search radius; $(x-x_i)^2 + (y-y_i)^2$ is the square of the distance between the $n$ *point* $x$, and $h$ is the sum of $(x_i, y_i)$ and $(x, y)$.

### 2.2.3. Exploratory Spatial Data Analysis (ESDA)

We next sought to describe and visualize the spatial correlation pattern of express delivery outlets via ESDA, as well as to identify abnormal agglomeration and distribution patterns. This approach is a collection of spatial data analysis methods and technologies. Through ESDA, the spatial correlation characteristics and patterns of China's express outlets were described and analyzed at both global and local scales. Specifically, the Global Moran's I index was used to measure the spatial correlation characteristics of express outlets on a national scale, after which Getis-Ord $G_i^*$ was used to analyze local hot and cold spots to identify high-value and low-value clusters at different locations. Clustering and outlier analysis (Anselin Local Moran's I) further identified high- and low-value clusters and outlier areas of express outlets.

(1) Global Moran's I Index

$$I = \frac{\sum_{i=1}^{n} \sum_{j=1}^{n} W_{ij}(x_i - \overline{x})(x_j - \overline{x})}{S^2 \sum_{i=1}^{n} \sum_{j=1}^{n} W_{ij}} (i \neq j) \tag{2}$$

In the formula: $n$ is the number of spatial units; $x_i$, $x_j$, and $\overline{x}$ are the $i$ and $j$ number of Chinese express outlets in the spatial unit and the average of all research units, respectively; $S^2$ is $\sum_{i=1}^{n}(x_i - \overline{x})^2/n$; $W_{ij}$ is the spatial weight matrix; spatial adjacency is 1 and non-adjacency is 0.

(2) Getis-Ord $G_i^*$

$$G_i^*(d) = \sum_{i=1}^{n} W_{ij}(d)x_j / \sum_{i=1}^{n} x_j \tag{3}$$

where $x_j$ is the $j$ number of express outlets in the spatial unit and $W_{ij}(d)$ is the space weight matrix.

(3) Clustering and outlier analysis (Anselin Local Moran's I)

Spatial clustering is an important analytical tool in spatial statistics, which divides spatial elements into categories according to their similarity. Clustering and outlier analysis (Anselin Local Moran's I), as a bottom-up systematic clustering method, compares elements with surrounding elements according to a certain attribute, calculates Moran's I index value and Z score, and thus spatially identifies significant high and low values and outliers [34].

2.2.4. Geodetector

Unlike conventional statistical methods, which make many assumptions, the geographic detector model makes few assumptions and can thus overcome the limitations of statistical methods regarding variables. Therefore, this approach is widely used in the study of the impact mechanisms of social and economic factors and natural environment factors [35]. The factor detection module of geographic detectors can identify influencing factors. The core principle of this factor detection module is that study objects always exist in specific spatial locations and the environmental factors that affect their development and changes have spatial variations. If these environmental factors and geographical locations are significantly consistent in space, this means that this environmental factor has a decisive effect on the occurrence and development of the study object [36].

## 3. Spatial Distribution Characteristics of Express Outlets in China

### 3.1. Distribution Characteristics of the Overall Pattern

Based on POI (Point of Interest) data, the standard deviation ellipse and kernel density methods were used to draw the spatial distribution pattern of express outlets in China (Figure 1). Notably, our findings indicated that:

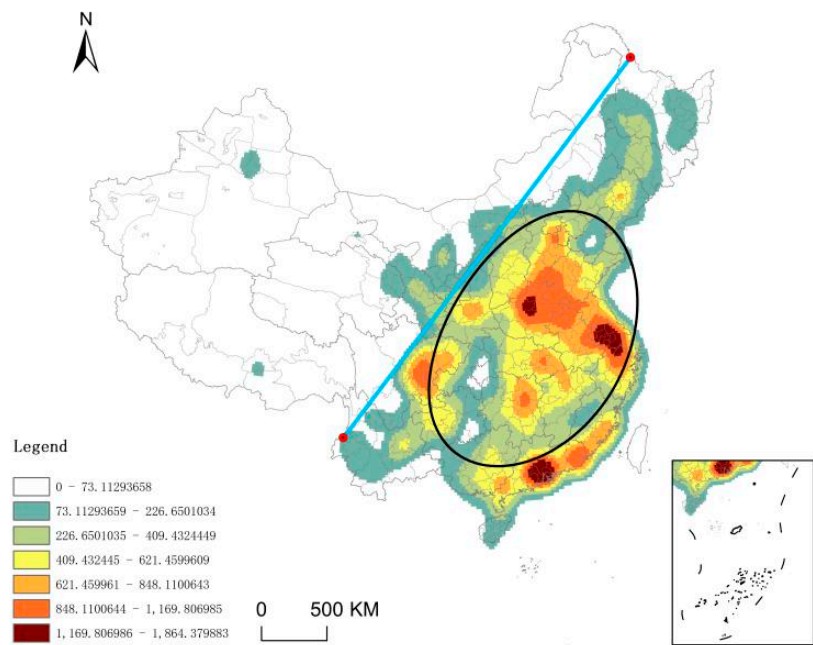

**Figure 1.** Kernel density map of the spatial distribution of express outlets in China. (Note: The figure is based on the standard map with the approval number of GS(2019)182 No. 2 on the Standard Map Service Website of the Ministry of Natural Resources; the boundaries of the base map have not been modified).

(1) China's express delivery outlets are generally more densely distributed in the east and more sparsely in the west. In the highly populated southeastern region, the distribution of express outlets is also very dense; in the sparsely populated northwest region, the express outlets are also sparsely distributed. At the same time, the standard deviation ellipse of express outlets presents a "northeast-southwest" trend, with the long-axis direction of the ellipse being "southwest-northeast" and the short-axis direction being "northwest-southeast," which is relatively parallel to the Hu Huanyong line (Hu Huanyong Line: The dividing line of population density in China. From Aigun, Heilongjiang Province to Tengchong, Yunnan Province, the land area to the east of the Hu Huanyong Line accounted for approximately 36% of the country, while the population at that time accounted for 96% of the country; the land area to the west accounted for 64% of the country but only contained 4% of the population). According to the data from China's sixth census, the population

density east of the Hu Huanyong line is 303.92 people/km$^2$, whereas the population density in the west is 15.72 people/km$^2$ [37].

(2) Furthermore, from the perspective of local density division, a "2 + 1 + 5" agglomeration pattern was observed. Two major agglomeration areas were observed: (1) the northern agglomeration area composed of the Central Plains urban agglomeration, the Shandong Peninsula urban agglomeration, and the Beijing-Tianjin-Hebei urban agglomeration, and (2) the eastern agglomeration area centered on the Yangtze River Delta urban agglomeration. Additionally, the Pearl River Delta urban agglomeration formed a belt-like agglomeration on the southeast coast. Finally, five point-like agglomeration areas were also observed, with Shenyang, Xi'an, Chengdu, Wuhan, and Changsha as their center points. The core density centers of several agglomeration areas were all economically developed or relatively developed cities. In 2020, the GDP of core cities in the agglomeration areas such as Shanghai, Beijing, Guangzhou, Chengdu, Wuhan, Changsha, Xi'an, and Shenyang will be RMB 3870.058 billion, RMB 3610.26 billion, RMB 2501.911 billion, RMB 1771.67 billion, RMB 1561.61 billion, RMB 1214.252 billion, RMB 1002.039 billion, and RMB 657.16 billion, respectively, ranking 1, 2, 4, 7, 10, 15, 22, and 33 in the country. The development of internal transportation and communication with a high level of informatization can lay a solid foundation for the development of the express delivery industry. With the further development of e-commerce, the State Council issued the "Opinions on Promoting the Coordinated Development of E-commerce and Express Logistics" in 2018, meaning that the requirements for the express industry are becoming progressively higher, and therefore the continued development of the express industry is still a priority. Therefore, the distribution and development direction of express outlets will be further optimized and improved in the future.

### 3.2. Spatial Correlation Distribution Characteristics

Moran's I index is 0.71 under a 1% significance test level, indicating that express delivery outlets were obviously clustered in space. According to the agglomeration theory, social and economic activities will generally be concentrated in a certain area, and related industries will also gather in a certain area, forming an agglomeration effect under the interaction of external effects and industrial scale [38,39]. Wilson (1981) proposed that agglomeration also benefits express delivery companies. Express delivery companies provide high-quality, fast, and efficient services to customers in the area by building a large number of outlets in the agglomeration area [40]. The distribution of outlets by express companies in a certain spatial gathering area is conducive to generating economies of scale and increasing the income of express companies. At the same time, the nearest neighbor index of national express outlets was calculated to be 0.2041, which indicates that the actual nearest neighbor distance between express outlets was smaller than the theoretical nearest neighbor distance, indicating that China's express outlets are clustered in space. Under a 99% confidence interval, the p-value is 0.0000 but the Z value is −645.6176, which is much smaller than −2.58, indicating that there is a very obvious clustering in the spatial distribution of express outlets. Therefore, our study further explored their distribution through hot spot analysis tools.

### 3.3. Spatial Hierarchical Structure Features

We next calculated the local correlation index GetisOrd $G_i^*$ of express outlets at a city scale and applied the Jenks natural breakpoint method to divide its spatial distribution into hot spots, sub-hot spots, sub-cold spots, and four types of cold spots. Figure 2 illustrates the spatial distribution of express delivery outlet hot spots in China.

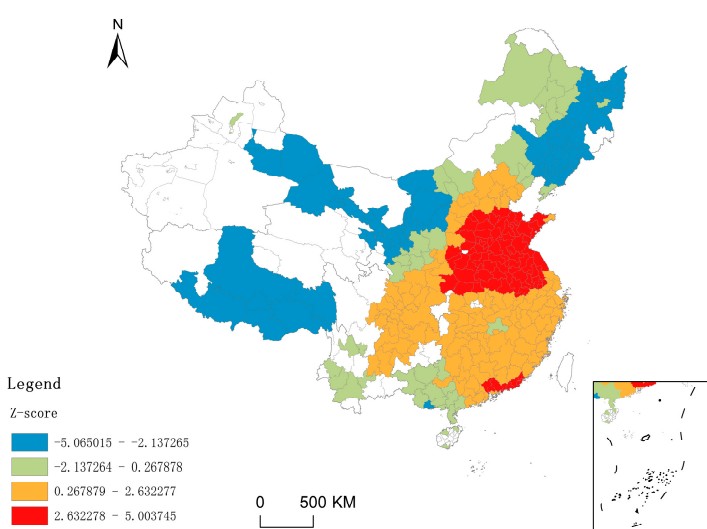

**Figure 2.** Spatial distribution of express outlet hotspots in China.

The data was then analyzed based on the principle of supply and demand balance. The express delivery market must maintain a stable and healthy development. Therefore, express delivery companies should follow the principle of supply and demand balance when deploying outlets. In this context, "supply" is the service provided by express companies to customers, whereas "demand" is the consumer's desire to acquire express delivery services within a certain time and range. Express delivery companies make the express delivery market reach a relatively balanced state within a certain area by reasonably adjusting the number of outlets and service scope [41]. Based on these criteria, a certain hierarchical structure is formed in space.

The hot and cold spots of express outlets exhibited a concentric structure, with the hot spots at the center and encompassing some cities in the provinces of Shandong, Hebei, Henan, Shanxi, Hubei, Anhui, and Jiangsu, which are generally located in the central and northern parts of China, as well as Guangzhou and Shenzhen in Guangdong Province. The outer layer contained the sub-hot spots, and these cities account for the largest proportion. Sub-cold spots and cold spots were mainly distributed in the northwest, northeast, northwest, and southwest fringe areas of China. The overall development of these cities is relatively similar, and they are all relatively underdeveloped. Compared with a previous article, Shenyang was now classified as a high-value area of express delivery density. However, this region is located in the northernmost region of China. This region is relatively remote and the winter lasts for 5–6 months, the temperature is low, and the average winter temperature reaches −20 °C. Therefore, some product manufacturers do not provide free shipping or even exclude this region from their service scope altogether. Furthermore, the central cities in the region have a weak driving effect, unbalanced urban development, and underdeveloped infrastructure, all of which adds to the reasons why there are relatively few express delivery outlets in this region. The northwest and southwest regions have low population density and are lagging in social and economic development. Additionally, the regional terrain is complex, featuring mountains, plateaus, and deserts. This makes transportation rather difficult, resulting in the formation of obvious low-value clusters of express delivery outlets in the region.

### 3.4. Spatial Differentiation of Express Outlets

Clustering and outlier analysis (Anselin Local Moran's I) is a bottom-up systematic clustering method. Our study compared the number of express outlets with the surrounding express outlets to identify significant high- and low-value areas, as well as outliers. The following conclusions can be drawn from Figure 3:

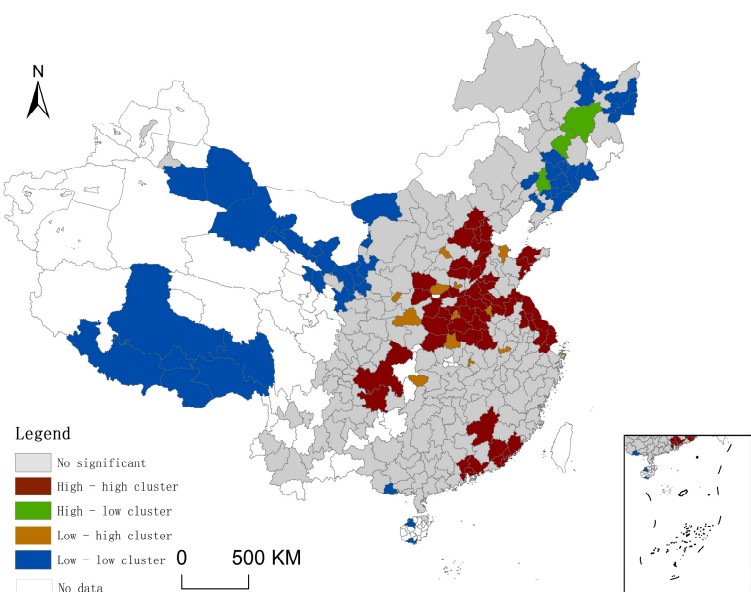

**Figure 3.** Spatial clustering and outlier distribution of express outlets in China. (Note: The figure is based on the standard map with the approval number of GS(2019)182 No. 2 on the Standard Map Service Website of the Ministry of Natural Resources, the boundaries of the base map have not been modified).

(1) Harbin, Changchun, and Shenyang in Northeast China are high-low clusters, that is, the number of express outlets in these three cities is higher than the number of express outlets in surrounding cities. This is because Harbin, Changchun, and Shenyang are the three major growth poles for the development of the Northeast. As such, these cities are developing in various aspects such as the tertiary industry, economic construction, transportation infrastructure construction, and informatization development, all of which are closely related to the express delivery industry. Most are ahead of other cities, which explains our observations.

(2) The outliers of the "low-high" cluster are scattered in the high-value area, and the distribution is relatively scattered. However, the reasons for this result are not the same. In the previous analysis, this area was identified as a hot spot. In this region, the overall number of express outlets is relatively large. However, due to the development of individual cities or natural factors, the number of express delivery outlets is small, resulting in this clustering result. For example, Zhangjiajie City is located in the hinterland of the Wuling Mountains, which exhibits relatively complex terrain and a tourist city with a small resident population. Therefore, the demand for express delivery is low and the foundation to support the development of the express delivery industry is poor. In turn, the number of express delivery outlets is relatively small.

## 4. Factors Affecting the Spatial Distribution Pattern of China's Express Delivery Outlets

### 4.1. Detection of Elements That Affect Express Outlets

The algorithm of the geographic detectors performs better for the analysis of categorical data than continuous data [35]. First, the ArcGIS software was used to cluster the continuous detection factor values by using the natural discontinuous point method, which is divided into categories 1, 2, 3, 4, and 5. The category space distribution of each detection factor is shown in Figure 4. Then, using the geographic detector measurement method, the PD and G values that reflect the influence of each detection factor on the distribution of express outlets are calculated from the perspectives of the whole country and the east, middle, and west regions (Table 1). The geodetector analysis results indicated the following:

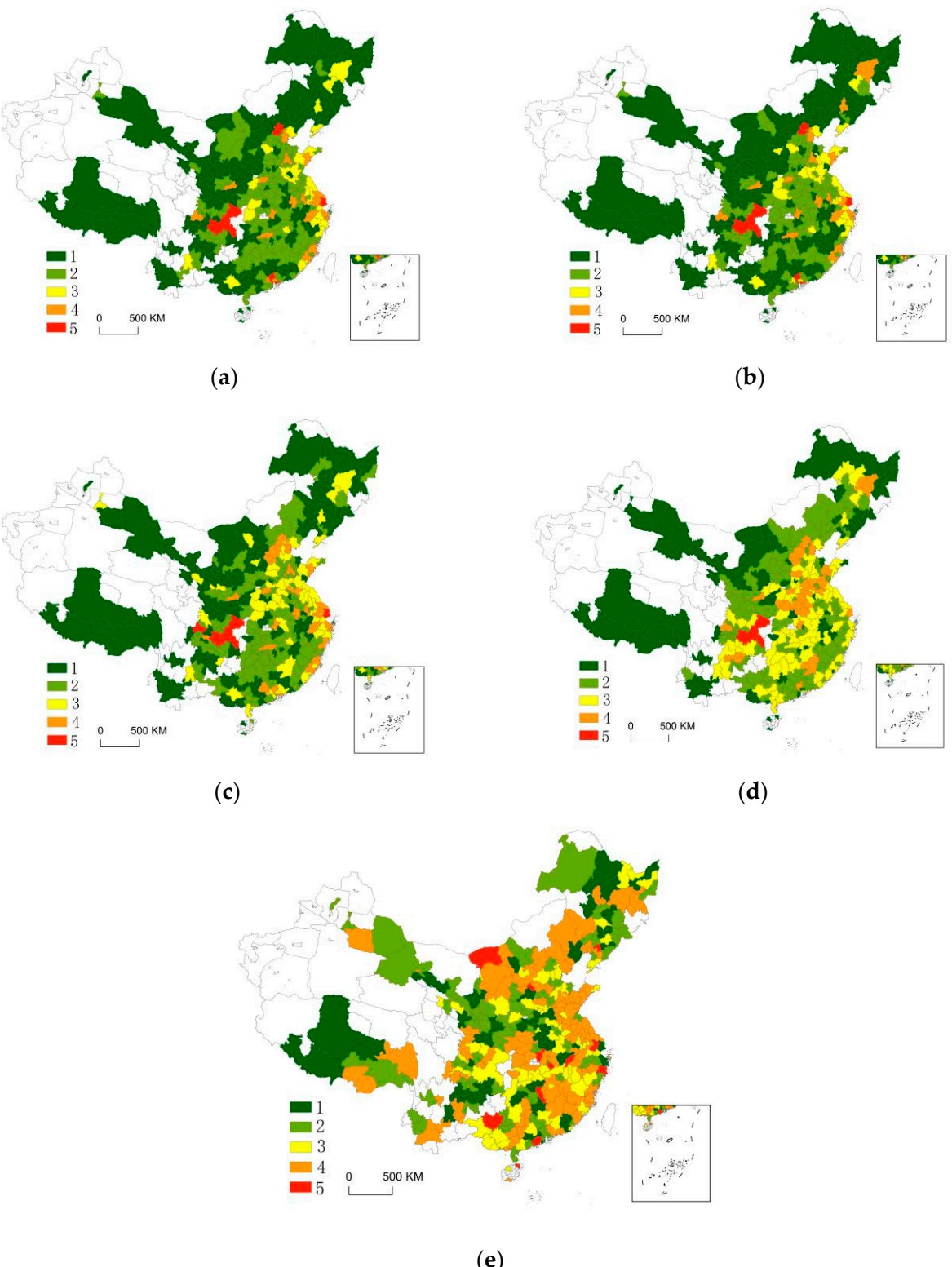

**Figure 4.** Categorical spatial distribution of geographic detection factors. (**a**) GDP. (**b**) Total social retail sales. (**c**) Number of internet users. (**d**) Population. (**e**) Road network density. (Note: The figure is based on the standard map with the approval number of GS(2019)182 No. 2 on the Standard Map Service Website of the Ministry of Natural Resources, the boundaries of the base map have not been modified).

**Table 1.** Detection results of the influencing factors of the distribution pattern of express outlets.

| Detection Indicators | PD, G | | | |
|---|---|---|---|---|
| | National | West | Central | East |
| GDP | 0.4483 | 0.3977 | 0.3727 | 0.4849 |
| Total social retail sales | 0.5244 | 0.4205 | 0.4923 | 0.5637 |
| Number of internet users | 0.6290 | 0.5352 | 0.6460 | 0.6585 |
| Population | 0.6897 | 0.7609 | 0.7706 | 0.5083 |
| Road network density | 0.0089 | 0.0295 | 0.0516 | 0.0232 |

(1) From the perspective of detection factors, the influence of each factor on the distribution of express outlets in different partitions shows specific consistency and differences. First of all, the performance of GDP, total retail sales, and the number of internet users are relatively consistent, indicating that from a national and a regional perspective, the level of economic development, the size of the consumer demand, and the scope of internet penetration have little effect on the distribution of the express outlets. The population size in the central and western regions has a high significance on the distribution of express outlets. Furthermore, the impact of road network density on the distribution of express outlets is low because the areas with high road network density have more convenient transportation, thus enhancing the service scope of express outlets. Moreover, cities will develop further, which will decrease the distribution of outlets in this area. Additionally, some cities will increase the proportion of industrial, commercial, residential, and other land uses during the construction process, resulting in a lower road network density, all of which is critical for express delivery. The demand and supply of express delivery are still considerably high, meaning that the road network density is less decisive for the distribution of express outlets.

(2) The PD, G values of each detection factor are not much different, and there is no particularly significant decisive factor. From the perspective of factor detection by partition, the decisive role of specific detection factors on the distribution of express outlets has become prominent. The reason for this result is that the factors affecting the distribution of express outlets within each region are similar, and the regional conditions and development conditions in China are different, resulting in large regional differences in detection factors. Therefore, the distribution of express outlets in various regions exhibits similar development conditions, development levels, and development trends.

(3) From a regional perspective, the prominent factors affecting the distribution of express outlets are different. The prominent factors affecting the distribution pattern of express outlets in the central and western regions are mainly population factors (0.7609 and 0.7706), whereas the main factor affecting the distribution pattern of express outlets in the eastern region is the number of internet users, which can be regarded as an indicator of technological development. However, in general, the PD, G values of all detection factors within each region are not much different (Table 1), which also shows that the distribution of express outlets is the result of the combined action of multiple factors, and effective policy measures are needed to promote its further development.

*4.2. Theoretical Analysis of Decisive Influencing Factors*

The results of our geographic exploration analysis revealed the influencing factors affecting the distribution of express outlets and their differences in the three major regions.

(1) Level of economic development

GDP is an important factor affecting the development of the express delivery industry in various cities. After the reform and opening-up period, cities in eastern China developed first and more rapidly. The economy has a great impact on infrastructure construction, the demand and supply of express delivery businesses, and the profitability of express delivery companies. Particularly, GDP has the most significant impact on the eastern region compared with the three major regions. Furthermore, the western region of China

is relatively lagging in terms of economic development. During the development of the express delivery industry, the industry has gradually raised the entry threshold, including economic requirements. Therefore, compared with the central region, the impact of economic conditions is more significant.

(2)　Social purchasing power

The total retail sales of social consumer goods reflect the improvement of people's material and cultural living standards in a certain period, as well as the resident's purchasing power and the size of the retail market. Approximately 60–80% of the orders of China's current express delivery companies come from e-commerce websites [17], and most of these Chinese e-commerce companies are located in the eastern and central regions, which drive local economic development and the purchasing power of the residents of these regions, thus affecting the network distribution of express delivery outlets.

(3)　Information technology development level

The rapid development of the express delivery industry is partly due to the accelerated development of information technology. The reasons for the close relationship between the two are obvious: express delivery outlets mainly accept e-commerce parcels. Therefore, cities with a large number of internet users and advanced information technology will have higher business volumes. The larger the number of express delivery outlets, the greater the number of layout express outlets. Central and eastern China have a high degree of informatization, and developed cities are particularly located in the eastern region. The development of eastern cities has gradually entered a new stage of smart city and informatization development. Therefore, the level of informatization has a prominent influence on the distribution of express outlets. Therefore, the future development of express outlets will be more dependent on the development of information technology.

(4)　Population

China is generally more densely populated in the east and less in the west. In the western region, not only is the population distribution small, but the population distribution is also relatively concentrated due to topographical reasons. The rapid development of the express industry, unmanned delivery, the use of Fengchao express cabinets, and the development of rookie stations all affect the distribution of express outlets to varying degrees. As the forefront of China's development in the eastern region, major cities are the first to reform and experience the advantages of technological development. However, investment in new technologies remains slow in the central and western regions, and therefore the population is still an important factor in determining the distribution of express outlets.

## 5. Conclusions

Based on the nearest neighbor index, kernel density, standard deviation ellipse, ESDA, and other spatial classification methods, the overall pattern characteristics and regional differences of China's express delivery outlets were systematically identified. Using geographic factor detection, the factors affecting the distribution of China's express delivery outlets were objectively analyzed. The following are the main conclusions of our study:

(1)　From a national perspective, express delivery outlets tended to be more abundant in the east and sparser in the west, and the distribution of express outlets is clustered forming a northern agglomeration area consisting of the Central Plains urban agglomeration, the Shandong Peninsula urban agglomeration, and the Beijing-Tianjin-Hebei urban agglomeration. The Yangtze River Delta urban agglomeration consists of the eastern agglomeration area in the center, the southeast coastal belt-like agglomeration area centered on the Pearl River Delta urban agglomeration, and the point-like agglomeration area centered on Shenyang, Xi'an, Chengdu, Wuhan, and Changsha, indicating that express delivery outlets can be easily developed in economically

developed areas, with internal traffic communication among cities and high levels of informatization.

(2) Regarding the local agglomeration of express outlets: seven cities in central and northern China, Guangzhou, Shenzhen, and other seven cities in Guangdong Province were classified as hot spots. Due to low demand, there is an obvious low-value cluster of express delivery outlets in the northwest, northeast, and southeast fringes. Additionally, outlier clustering was observed in the northeastern region and the central recipient region.

(3) The spatial pattern of China's express delivery outlets is affected by factors such as population, economic development, information technology, and residents' purchasing power. The decisive factors affecting the distribution of express delivery outlets vary depending on the region. Faced with this problem, effective policy measures need to be implemented to promote the further development of the express delivery industry.

## 6. Discussion

The conclusions of this paper provide key insights into the overall pattern and influencing factors of China's express delivery network distribution. However, there are still some topics that have not been properly explored due to space and data limitations.

(1) There is a huge number of express delivery outlets in China. Moreover, due to the vast territory of China, the development of different regions and units at different administrative levels is relatively strong. Therefore, future research should focus on comparing and analyzing the distribution of express outlets in different administrative units in China to explore the reasons for their formation.

(2) In recent years, after Alibaba established the Cainiao Network, the self-pickup cabinets and couriers established by the Cainiao Network have become the final link of express delivery to consumers, which has had a considerable impact on the express delivery industry. Therefore, future studies must compare and/or comprehensively analyze these networks.

(3) The distribution of express delivery outlets is the result of multiple factors (e.g., socioeconomic and environmental factors), as well as institutional policies. Moreover, express delivery outlets in China mostly exhibit corporate behaviors. Therefore, future studies must incorporate more auxiliary databases such as special investigations. Additionally, our findings suggest that the influence of traditional factors on the distribution of express delivery outlets (e.g., demographic factors) will gradually decrease under the rapid development of express delivery industry-related fields. Instead, new technologies will act as driving forces or even determine the patterns of express delivery outlets and new dynamics of spatial distribution.

**Author Contributions:** Conceptualization, X.L. and P.Z.; methodology, X.L.; software, X.L.; validation, X.L.; formal analysis, X.L.; investigation, X.L.; resources, P.Z.; data curation, X.L.; writing—original draft preparation, X.L.; writing—review and editing, X.L.; visualization, X.L.; supervision, P.Z.; project administration, X.L.; funding acquisition, P.Z. All authors have read and agreed to the published version of the manuscript.

**Funding:** This research was funded by the Heilongjiang Province General Undergraduate High School Youth Innovation Talent Training Program Project (No. UN-PYSCT-2017193), the Harbin Normal University Ph.D. Startup Fund Project (No. XKB201815), the Harbin Normal University Graduate Student Innovation Research Program (HSDSSCX2021-25).

**Institutional Review Board Statement:** Not applicable.

**Informed Consent Statement:** Not applicable.

**Data Availability Statement:** The express outlets were obtained from the AutoNavi map open platform by calling the web service API interface (https://lbs.amap.com/api/webservice/guide/api/newpoisearch).

**Conflicts of Interest:** The authors declare no conflict of interest.

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
