# Peer review of "Patterns and Influencing Factors of Express Outlets in China"

_sustainability, doi:10.3390/su14138061_

Round 1

Reviewer 2 Report

After detailed reading, the paper is relevant and interesting. This paper fits to this special issue.

The main question addressed by the research is the Patterns and influencing factors of express outlets in China.

The authors have offered some new approaches that correspond to the main question of the research – factors of express outlets in China.

A new perspective towards patterns and influencing factors of express outlets are offered throughout the Chinese experience.

The conclusion is consistent with the presented arguments and evidence.

Round 2

Reviewer 1 Report

Dear Authors, 

Please read my comments carefully. I have accessed your revision version but some comments were not addressed appropriately. Therefore I would like to invite you for another revision chance. 

Moreover, please pay attention to langue quality in this paper. I have found numerous grammatical mistake. I suggest that authors must be required to do proofreading. 

Finally, check the whole manuscript, the plenty of statement does not elaborate  updated literature. Personal opinions have no scholarly documents. 

Round 3

Reviewer 1 Report

Thanks for your revision.

Generally, I am satisfied with responses to my comments. 

All the best for your research.